# Asymmetric Temperature Scaling Makes Larger Networks Teach Well Again

**Xin-Chun Li**[1], **Wen-Shu Fan**[1], **Shaoming Song**[2], **Yinchuan Li**[2]
**Bingshuai Li**[2], **Yunfeng Shao**[2], **De-Chuan Zhan**[1]
[1] State Key Laboratory for Novel Software Technology, Nanjing University, Nanjing, China
[2] Huawei Noah's Ark Lab, Beijing, China
{lixc, fanws}@lamda.nju.edu.cn, zhandc@nju.edu.cn
{shaoming.song, liyinchuan, libingshuai, shaoyunfeng}@huawei.com

## Abstract

Knowledge Distillation (KD) aims at transferring the knowledge of a well-performed neural network (the *teacher*) to a weaker one (the *student*). A peculiar phenomenon is that a more accurate model doesn't necessarily teach better, and temperature adjustment can neither alleviate the mismatched capacity. To explain this, we decompose the efficacy of KD into three parts: *correct guidance*, *smooth regularization*, and *class discriminability*. The last term describes the distinctness of *wrong class probabilities* that the teacher provides in KD. Complex teachers tend to be over-confident and traditional temperature scaling limits the efficacy of *class discriminability*, resulting in less discriminative wrong class probabilities. Therefore, we propose *Asymmetric Temperature Scaling (ATS)*, which separately applies a higher/lower temperature to the correct/wrong class. ATS enlarges the variance of wrong class probabilities in the teacher's label and makes the students grasp the absolute affinities of wrong classes to the target class as discriminative as possible. Both theoretical analysis and extensive experimental results demonstrate the effectiveness of ATS. The demo developed in Mindspore is available at https://gitee.com/lxcnju/ats-mindspore and will be available at https://gitee.com/mindspore/models/tree/master/research/cv/ats.

## 1 Introduction

Although large-scale deep neural networks have achieved overwhelming successes in many real-world applications [22, 11, 60], the vast capacity hinders them from being deployed on portable devices with limited computation and storage resources [3]. Some efficient architectures, e.g., MobileNets [14, 37] and ShuffleNets [59, 29], have been proposed for lightweight deployment, while their performances are usually constrained. Fortunately, knowledge distillation (KD) [46, 13] could transfer the knowledge of a more complex and well-performed network (i.e., the *teacher*) to them.

The original KD [13] forces the student to mimic the teacher's behavior via minimizing the Kullback-Leibler (KL) divergence between their output probabilities. Recent studies generalize KD to various types of knowledge [36, 57, 17, 12, 33, 1, 34, 44, 52, 27, 54, 45, 50, 26, 23] or various distillation schemes [61, 2, 58, 20]. An intuitive sense after the proposal of KD [13] is that larger teachers could teach students better because their accuracies are higher. A recent work [6] first points out that the teacher accuracy is a poor predictor of the student's performance. That is, more accurate neural networks don't necessarily teach better. Until now, this phenomenon is still counter-intuitive [51], surprising [31], and unexplored [24]. Different from some existing empirical studies and theoretical analysis [40, 18, 30, 35, 55, 63, 6, 28, 15], we investigate the miraculous phenomenon in detail and aim to answer the following questions: *What's the real reason that more complex teachers can't teach*

36th Conference on Neural Information Processing Systems (NeurIPS 2022).

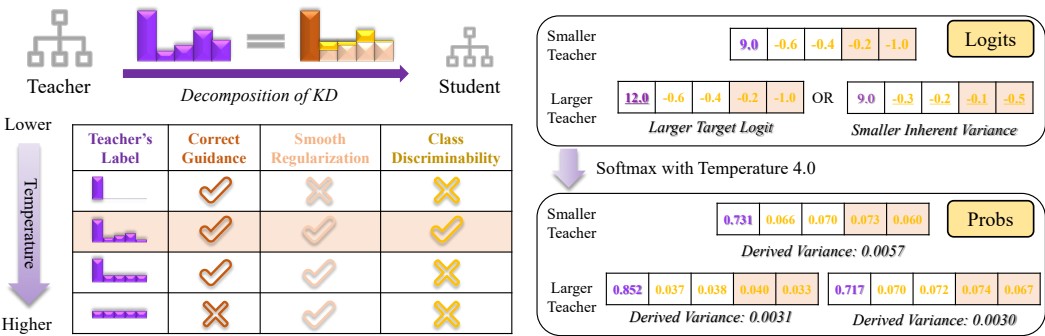

Figure 1: **Left**: Decomposition of a teacher's label. The first class is the target. As temperature increases, *correct guidance* is weaker, *smooth regularization* is stronger, while *class discriminability* (measured by the variance of wrong class probabilities) will first increase and then decrease. **Right**: Larger/Smaller teachers' logits are consistent in relative class affinities, i.e., logit values of the four wrong classes are in the same order of magnitude. However, larger teachers are over-confident and give a larger target logit or smaller *inherent variance*, leading to a smaller *derived variance* under traditional temperature scaling, i.e., less distinct wrong class probabilities after softmax.

*well? Is it really impossible to make larger teachers teach better through simple operations, such as temperature scaling?*

To answer the first question, we focus on analyzing the distinctness of wrong class probabilities that a teacher provides in KD. We decompose the teacher's label into three parts (see Sect. 4.1): (I) *Correct Guidance*: the correct class's probability; (II) *Smooth Regularization*: the average probability of wrong classes; (III) *Class Discriminability*: the variance of wrong class probabilities (defined as *derived variance*). The commonly utilized temperature scaling could control the efficacy of these three terms (the left of Fig. 1). More complex teachers are over-confident and assign a larger score for the correct class or less varied scores for the wrong classes. If we use a uniform temperature to scale their logits, the *class discriminability* of the larger teacher is less effective (theoretically analyzed in Sect. 4.2), i.e., the probabilities of wrong classes are less distinct (the right of Fig. 1).

As to the second question, we focus on enlarging the variance of wrong class probabilities (i.e., *derived variance*) that a teacher provides to make the distillation process more discriminative. To specifically enhance the distinctness of wrong class probabilities, we separately apply a higher/lower temperature to the correct/wrong class's logit instead of a uniform temperature (see Sect. 4.3). We name our method *Asymmetric Temperature Scaling (ATS)*, and abundant experimental studies verify that utilizing this simple operation could make larger teachers teach well again.

## 2   Related Works

**KD with Larger Teacher**: Although KD has been a general technique for knowledge transfer in various applications [13, 61, 42, 25], could any student learn from any teacher? [6] first studies the KD's dependence on student and teacher architectures. They find that larger models do not often make better teachers and propose the *early-stopped teacher* as a solution. [31] introduces a multi-step KD process, employing an intermediate-sized network (the *teacher assistant*) to bridge the capacity gap. [51] formulates KD as a multi-task learning problem with several knowledge transfer losses. The transfer loss will be utilized only when its gradient direction is consistent with the cross-entropy loss. [10, 24] define the knowledge gap as *residual*, which is utilized to teach the *residual student*, and then they take the ensemble of the student and residual student for inference. These works attribute the worse teaching performance to capacity mismatch, i.e., weaker students can't completely mimic the excellent teachers. However, they don't explain this peculiar phenomenon in detail.

**Understanding of KD**: Quite a few works focus on understanding the advantages of KD from a principled perspective. [28] unifies KD and privileged information into *generalized distillation*. [35, 18] utilize gradient flow and neural tangent kernel to analyze the convergence property of KD under deep linear networks and infinitely wide networks. [5] explains KD via quantifying the task-

Table 1: The used notations in this paper. The definitions of *Derived Average*, *Derived Variance* and *Inherent Variance* are only for wrong classes (Sect. 4.1 and Sect. 4.2).

| | All Classes | Wrong Classes |
|---|---|---|
| Logit | $\mathbf{f}$ | $\mathbf{g} = [\mathbf{f}_c]_{c \neq y}$ |
| Probability | $\mathbf{p} = \text{SF}(\mathbf{f})$ | $\mathbf{q} = [\mathbf{p}_c]_{c \neq y}$ |
| Derived Average of Probabilities | - | $e(\mathbf{q}) = \sum_j \mathbf{q}_j / (C-1)$ |
| Derived Variance of Probabilities | - | $v(\mathbf{q}) = \sum_j (\mathbf{q}_j - e(\mathbf{q}))^2 / (C-1)$ |
| Inherent Variance of Probabilities | - | $\tilde{\mathbf{q}} = \text{SF}(\mathbf{g}), v(\tilde{\mathbf{q}}) = \sum_j (\tilde{\mathbf{q}}_j - e(\tilde{\mathbf{q}}))^2 / (C-1)$ |

relevant and task-irrelevant visual concepts. [7] casts KD as a semiparametric inference problem and proposes corresponding enhancements. Our work is more related to KD decompositions. [9] treats the teacher's correct/wrong outputs differently, respectively explaining them as importance weighting and class similarities. [40] further decomposes the "dark knowledge" into universal knowledge, domain knowledge, and gradient rescaling. [30] establishes a bias-variance tradeoff to quantify the divergence of a teacher with the *Bayes teacher*. [63] utilizes bias-variance decomposition to analyze KD and discovers *regularization samples* that could increase bias and decrease variance. Our work is also related to Label smoothing (LS). [55] points out that the regularization effect in KD is similar to LS. [32] finds that training a teacher with LS could degrade its teaching quality, and attributes this to the fact that LS erases *relative information* between teacher logits. Recently, [38] further studies this problem and proposes a metric to measure the degree of erased information quantitatively. Our work also decomposes KD into several effects to study why more complex teachers can't teach well. Detailed relatedness to these works is presented in Sect. 4.1.

## 3 Background

We consider a $C$-class classification problem with $\mathcal{Y} = [C] = \{1, 2, \cdots, C\}$. Given a neural network and a sample pair $(\mathbf{x}, y)$, we could obtain the "logits" as $\mathbf{f}(\mathbf{x}) \in \mathbb{R}^C$. We denote the softmax function with temperature $\tau$ as $\text{SF}(\cdot; \tau)$, i.e., $\mathbf{p}_c(\tau) = \exp(\mathbf{f}_c(\mathbf{x})/\tau) / Z(\tau)$ and $Z(\tau) = \sum_{j=1}^C \exp(\mathbf{f}_j(\mathbf{x})/\tau)$, where $\mathbf{p}(\tau)$ is the softened probability vector that a network outputs and $c$ is the index of class. Later, we may omit the dependence on $\mathbf{x}$ and $\tau$ if without any ambiguity. We use $\mathbf{f}_y$ and $\mathbf{p}_y$ to denote the *correct class*'s logit and probability, while we use $\mathbf{g}$ and $\mathbf{q}$ to represent the vector of *wrong classes*' logits and probabilities, i.e., $\mathbf{g} = [\mathbf{f}_c]_{c \neq y}$ and $\mathbf{q} = [\mathbf{p}_c]_{c \neq y}$. The notations could be found in Tab. 1.

The most standard KD [13] contains two stages of training. The first stage trains complex teachers, and then the second stage transfers the knowledge from teachers to a smaller student via minimizing the KL divergence between softened probabilities. Usually, the loss function during the second stage (i.e., the student's learning objective) is a combination of cross-entropy loss and distillation loss:

$$\ell = \underbrace{-(1-\lambda)\log \mathbf{p}_y^S(1)}_{\text{CE Loss}} \underbrace{-\lambda\tau^2 \sum_{c=1}^C \mathbf{p}_c^T(\tau)\log \mathbf{p}_c^S(\tau)}_{\text{KD Loss}}, \tag{1}$$

where the upper script "T"/"S" denotes "Teacher"/"Student" respectively. Commonly, a default temperature of $1$ is utilized for the CE loss, and the student could also take a temperature of $1$ for the KD loss, e.g., $\mathbf{p}_c^S(\tau = 1)$ [13, 31, 45, 44].

Suppose we have two teachers, denoted as $T_{\text{large}}$ and $T_{\text{small}}$, and the larger teacher performs better on both training and test data. If we use them to teach the same student $S$, we could find that the student's performance is worse when mimicking the larger teacher's outputs. Adjusting the temperature could neither make the larger teacher teach well. The details of this phenomenon could be found in [6, 31] and Fig. 9. Obviously, $\mathbf{p}^{T_{\text{large}}}$ could differ a lot from $\mathbf{p}^{T_{\text{small}}}$, which is the only difference in the loss function when teaching the student. Hence, we focus on analyzing *what probability distributions are tended to be provided by teachers with different capacities*.

# 4 Proposed Methods

This section first decomposes KD into three parts and defines several quantitative metrics. Then, we present theoretical analysis to demonstrate why larger networks can't teach well. Finally, we propose a more appropriate temperature scaling approach as an alternative.

## 4.1 KD Decomposition

We omit the coefficient of $\lambda \tau^2$ in Eq. 1, and define $e\left(\mathbf{q}^T(\tau)\right) = \frac{1}{C-1} \sum_{j=1, j \neq y}^{C} \mathbf{p}_j^T(\tau)$, where $\mathbf{q}^T(\tau) = \left[\mathbf{p}_c^T(\tau)\right]_{c \neq y}$. Then, we have the following decomposition:

$$\ell_{\text{kd}} = \underbrace{-\mathbf{p}_y^T(\tau) \log \mathbf{p}_y^S(\tau)}_{\text{Correct Guidance}} - \underbrace{\sum_{c \neq y} e\left(\mathbf{q}^T(\tau)\right) \log \mathbf{p}_c^S(\tau)}_{\text{Smooth Regularization}} - \underbrace{\sum_{c \neq y} \left(\mathbf{p}_c^T(\tau) - e\left(\mathbf{q}^T(\tau)\right)\right) \log \mathbf{p}_c^S(\tau)}_{\text{Class Discriminability}}. \quad (2)$$

**(I) Correct Guidance**: this term guarantees correctness during teaching. The decomposition in [9] also contains this term, which is explained as importance weighting. This term works similarly to the cross-entropy loss, which could be dealt with separately when applying temperature scaling.

**(II) Smooth Regularization**: some previous works [55, 62, 63] attribute the success of KD to the efficacy of regularization and study its relation to label smoothing (LS). The combination of this term with *correct guidance* works similarly to LS. Notably, $e\left(\mathbf{q}^T(\tau)\right)$ differs across samples, implying that the strength of smoothing is instance-specific, which is similar to the analysis in [62].

**(III) Class Discriminability**: this term tells the student the affinity of wrong classes to the correct class. Transferring the knowledge of class similarities to students has been the mainstream guess of the "dark knowledge" in KD [13, 38]. Ideally, a good teacher should be as discriminating as possible in telling students which classes are more related to the correct class.

Illustrations of the decomposition are presented in the left of Fig. 1. Obviously, an appropriate temperature should simultaneously contain the efficacy of the three terms, e.g., the shaded row in Fig. 1. A too high or too low temperature could lead to smaller *class discriminability*, making the guidance less different among wrong classes, which weakens the distillation performance in practical. Among these three terms, we advocate that *class discriminability* is more fundamental in KD and present more discussions in Appendix A (verified in Fig. 2 and Fig. 3).

To measure these three terms quantitatively, we use the *target class probability* (i.e., $\mathbf{p}_y$), the *average of wrong class probabilities* (i.e., $e(\mathbf{q}) = \frac{1}{C-1} \sum_{j \neq y} \mathbf{p}_j$), and the *variance of wrong class probabilities* (i.e., $v(\mathbf{q}) = \frac{1}{C-1} \sum_{j \neq y} \left(\mathbf{p}_j - e(\mathbf{q})\right)^2$) as estimators. $e(\cdot)$ and $v(\cdot)$ calculates the mean and variance of the elements in a vector. In some cases, we use the standard deviation as an estimator for the third term, i.e., $\sigma(\mathbf{q}) = v^{1/2}(\mathbf{q})$. Because the latter two terms are calculated after applying softmax to the complete logit vector, we define them as *Derived Average (DA)* and *Derived Variance (DV)*, respectively. In experiments, we calculate these metrics for all training samples and sometimes report the average or standard deviation across these samples.

## 4.2 Theoretical Analysis

We analyze the mean and variance of the softened probability vector, i.e., the teacher's label $\mathbf{p}^T(\tau)$ used in KD. We defer the proofs of Lemma 4.1 and Proposition 4.3, 4.4 to Appendix B.

**Lemma 4.1** (Variance of Softened Probabilities). *Given a logit vector $\mathbf{f} \in \mathbb{R}^C$ and the softened probability vector $\mathbf{p} = SF(\mathbf{f}; \tau), \tau \in (0, \infty)$, $v(\mathbf{p})$ monotonically decreases as $\tau$ increases.*

As $\tau$ increases, $\mathbf{p}(\tau)$ becomes more uniform, i.e., it's entropy increases. However, we especially focus on the wrong classes, where the mean and variance are more intuitive to calculate and analyze.

**Assumption 4.2.** *The target logit is higher than other classes' logits, i.e., $\mathbf{f}_y \geq \mathbf{f}_c, \forall c \neq y$.*

Assumption 4.2 is rational because well-performed teachers could almost achieve a higher accuracy (e.g., >95%) on the training set, and most training samples meet this requirement.

**Proposition 4.3.** *Under Assumption 4.2, $\mathbf{p}_y$ monotonically decreases as $\tau$ increases, and $e(\mathbf{q})$ monotonically increases as $\tau$ increases. As $\tau \to \infty$, $e(\mathbf{q}) \to 1/C$.*

Proposition 4.3 implies that increasing temperature could lead to a higher *derived average* (empirically see Fig. 7) and strengthen the *smooth regularization* term in Eq. 2.

Before we analyze the *class discriminability* term, we define $\tilde{\mathbf{q}}(\tau)$ as the result of applying softmax *only to the wrong logits* with temperature $\tau$, i.e., $\tilde{\mathbf{q}}(\tau) = \mathrm{SF}(\mathbf{g}; \tau)$. For the element index $c'$ of $\mathbf{q}$, we have

$$\tilde{\mathbf{q}}_{c'}(\tau) = \exp(\mathbf{g}_{c'}/\tau) / \sum_j \exp(\mathbf{g}_j/\tau). \tag{3}$$

Notably, $\tilde{\mathbf{q}}$ differs from $\mathbf{q}$ a lot. Specifically, the former satisfies $\sum_{c'} \tilde{\mathbf{q}}_{c'} = 1$, while the summation of the latter is $\sum_{c \neq y} \mathbf{p}_c = 1 - \mathbf{p}_y$. The former does not depend on the correct class's logit while the latter does. We name $v(\tilde{\mathbf{q}})$ *Inherent Variance (IV)* because it only depends on wrong classes' logits.

**Proposition 4.4** (Derived Variance vs. Inherent Variance)**.** *The derived variance is determined by the square of derived average and the inherent variance via:*

$$\underbrace{v(\mathbf{q})}_{DV} = (C-1)^2 \underbrace{e^2(\mathbf{q})}_{DA^2} \underbrace{v(\tilde{\mathbf{q}})}_{IV}. \tag{4}$$

With $\tau$ increases, $e(\mathbf{q})$ increases (Proposition 4.3) while $v(\tilde{\mathbf{q}})$ decreases (Lemma 4.1), and hence, it is not so easy to judge the specific monotonicity of $v(\mathbf{q})$ w.r.t. $\tau$. Empirically, we observe that the *derived variance* first increases and then decreases (see Fig. 7), which conforms to the change of the *class discriminability* as illustrated in Fig. 1.

We could use Proposition 4.4 to clearly analyze why larger teacher networks can't teach well. Before this, we present another two properties and a corollary without detailed proof.

*Remark* 4.5. Fixing $\mathbf{g}$ and $\tau$, a higher target logit $\mathbf{f}_y$ leads to a higher $\mathbf{p}_y$, i.e., a smaller *derived average* $e(\mathbf{q})$.

*Remark* 4.6. Fixing $\tau$, less varied wrong logits $\mathbf{g}$ leads to less varied $\tilde{\mathbf{q}}$, i.e., a smaller *inherent variance* $v(\tilde{\mathbf{q}})$.

**Corollary 4.7.** *Suppose we have two teachers $T_1$ and $T_2$, and their logit vectors for a same sample are $\mathbf{f}^{T_1}$ and $\mathbf{f}^{T_2}$.*

- *If $\mathbf{f}_y^{T_1} \geq \mathbf{f}_y^{T_2}$ while $\mathbf{g}^{T_1}$ and $\mathbf{g}^{T_2}$ are nearly the same, then $\mathbf{p}_y^{T_1} \geq \mathbf{p}_y^{T_2}$ (Remark 4.5) while $v(\tilde{\mathbf{q}}^{T_1}) \approx v(\tilde{\mathbf{q}}^{T_2})$. Hence, $v(\mathbf{q}^{T_1}) \leq v(\mathbf{q}^{T_2})$.*

- *If $\mathbf{f}_y^{T_1} \approx \mathbf{f}_y^{T_2}$ while $v(\mathbf{g}^{T_1}) \leq v(\mathbf{g}^{T_2})$, then $\mathbf{p}_y^{T_1} \approx \mathbf{p}_y^{T_2}$ while $v(\tilde{\mathbf{q}}^{T_1}) \leq v(\tilde{\mathbf{q}}^{T_2})$ (Remark 4.6). Hence, $v(\mathbf{q}^{T_1}) \leq v(\mathbf{q}^{T_2})$.*

This corollary explains why a larger teacher can't teach better. Because the larger teacher tends to be over-confident, the target logit $\mathbf{f}_y$ may be larger or the variance of wrong logits $v(\mathbf{g})$ may be smaller. These are illustrated in Fig. 1 and empirically verified in Fig. 4. Then the *derived variance* $v(\mathbf{q})$ may be smaller, limiting the efficacy of *class discriminability* in Eq. 2. Empirical results are in Fig. 7.

Notably, we focus on analyzing the variance of wrong class probabilities instead of all classes. Maximizing the variance of all classes' probabilities does not mean maximizing the variance of wrong classes'. For example, although a very low temperature can maximize the variance of all classes' probabilities, the generated teacher's label is one-hot that shows no distinctness between wrong classes. In other words, the effectiveness of KD should be more related to the distinctness between wrong classes rather than all classes. However, traditional temperature scaling applies a uniform temperature for all classes, which cannot separately handle the wrong classes.

## 4.3 Asymmetric Temperature Scaling

We conclude the above analysis: *if a larger teacher makes an over-confident prediction, the wrong class probabilities it provides could be not discriminative enough.* Utilizing a uniform temperature could not enlarge the *derived variance* as much as possible with the interference of the target class's logit (see the middle of Fig. 7). Thanks to the decomposition in Eq. 2, the *correct guidance* term works similarly to the cross-entropy loss and allows us to deal with it separately. Hence, we propose a novel temperature scaling approach:

$$\mathbf{p}_c(\tau_1, \tau_2) = \exp\left(\mathbf{f}_c/\tau_c\right) / \sum_{j \in [C]} \exp\left(\mathbf{f}_j/\tau_j\right), \qquad \tau_i = \mathcal{I}\{i = y\}\tau_1 + \mathcal{I}\{i \neq y\}\tau_2, \forall i \in [C], \tag{5}$$

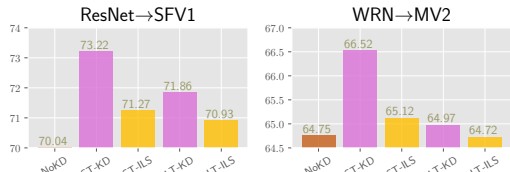
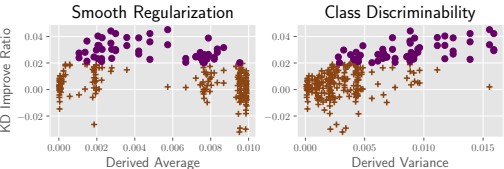

Figure 2: Student's test accuracies without KD ("NoKD"), with KD ("-KD"), and only with the first two terms in Eq. 2 ("-ILS"). "ST"/"LT" refers to "small/large teacher".

Figure 3: Correlations of *smooth regularization* (measured by *derived average*) and *class discriminability* (measured by *derived variance*) w.r.t. KD improvement ratio.

where we take $\tau_1 > \tau_2 > 0$. This approach is named *Asymmetric Temperature Scaling (ATS)* because we apply different temperatures to the logits of correct and wrong classes. According to Eq. 4, ATS could bring such benefits when the teacher is over-confident:

- If the teacher outputs a larger logit $\mathbf{f}_y$ for the correct class, a relatively larger $\tau_1$ could decrease it to a reasonable magnitude, i.e., decreasing $\mathbf{p}_y$ and increasing $e(\mathbf{q})$, and finally increasing the *derived variance* $v(\mathbf{q})$;

- If the teacher outputs less varied logits $\mathbf{g}$ for wrong classes, a relatively smaller temperature $\tau_2$ could make them more diverse, i.e., increasing $v(\tilde{\mathbf{q}})$, finally increasing the *derived variance* $v(\mathbf{q})$.

ATS is more flexible in enlarging the *derived variance* (see the right of Fig. 7), i.e., it could generate more discriminative distillation guidance during teaching. Take the demo in Fig. 1 as an example, the smaller and larger teacher captures the same relative class affinities, e.g., they both know that the fourth/fifth class (shaded cells) is the most/least relevant to the target class. However, with a uniform temperature $4.0$, the smaller teacher provides probabilities $(0.073, 0.060)$ for these two classes, while over-confident larger teachers provide $(0.040, 0.033)$ or $(0.074, 0.067)$. Clearly, the absolute affinities of the larger teachers are not so discriminative as the smaller teacher's. Utilizing ATS, we could respectively apply $(\tau_1 = 4.67, \tau_2 = 4.0)$ or $(\tau_1 = 4.0, \tau_2 = 2.0)$ to the over-confident teacher's logits, generating the same probability vector as the smaller teacher's. ATS utilizes two temperatures and creates the wiggle room to make the distribution over wrong classes more discriminative.

## 5 Experiments

We use CIFAR-10/CIFAR-100 [21], TinyImageNet [43], CUB [47], Stanford Dogs [19], and Google Speech Commands [48] as the datasets. For teacher networks, we use different versions of ResNet [11], WideResNet [56], ResNeXt [49]. For student networks, we use VGG [39], ShuffleNetV1/V2 [59, 29], AlexNet [22], MobileNetV2 [37], and DSCNN [60].

We majorly follow the training settings in [44] [1]. Except that the Google Speech Commands takes 50 epochs, we train networks on other datasets with 240 epochs. We use the SGD optimizer with 0.9 momentum. For VGG, AlexNet, ResNet, WideResNet, and ResNeXt, we set the learning rate as 0.05 (recommended by [44]). For ShuffleNet and MobileNet, we use a smaller learning rate of 0.01 (recommended by [44]). We use the pre-trained models provided in PyTorch for CUB and Stanford Dogs, and correspondingly, their learning rates are scaled by $0.1\times$. During training, we decay the learning rate by 0.1 every 30 epochs after the first 150 epochs (recommended by [44]). For Google Speech Commands, we decay the learning rate via the cosine annealing. We set the batch size as 128 for CIFAR data, 64 for other datasets. Other dataset, network and training details are in Appendix C.

### 5.1 Observations

**Class discriminability matters a lot in KD and correlates with the KD improvement.** To show the importance of *class discriminability* in KD, we omit the distinctness of wrong class probabilities during distillation. Specifically, we only keep the first two terms in Eq. 2, which works similarly to

---

[1] https://github.com/HobbitLong/RepDistiller



Figure 4: The distributions of the *target logit* ($\mathbf{f}_y$) and the *standard deviation of wrong logits* ($\sigma(\mathbf{g})$) of the 50K training samples on CIFAR-10/100. Rows show networks with various capacity.

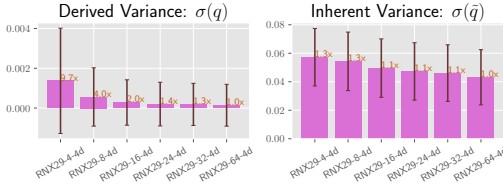

Figure 5: The *derived variance* and *inherent variance* on CIFAR-100 using ResNeXt. Each bar shows the mean and standard deviation across 50K training samples. The x-axis shows networks with various capacity.

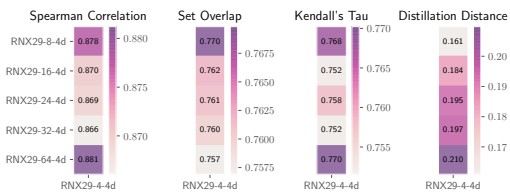

Figure 6: Several metrics among teachers with various capacities on CIFAR-10. The first three metrics are related to the relative magnitudes of teachers' label, while distillation distance is related to the absolute magnitudes.

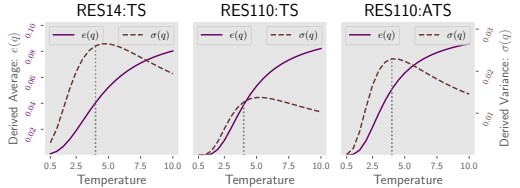

Figure 7: The change of *derived average* ($e(\mathbf{q})$) and *derived variance* ($v(\mathbf{q})$) as $\tau$ increases from 0.1 to 10.0 on CIFAR-10. The third one shows the results of ResNet110 with the proposed ATS. *DV* under TS is limited while ATS enlarges it.

the instance-specific label smoothing (abbreviated as "ILS"). Fig. 2 shows students' performances on CIFAR-100. Without the third term in Eq. 2, both small teachers ("ST") and larger teachers ("LT") teach worse significantly. Then, we investigate the correlations of KD performance improvement (i.e., the test accuracy change ratio with KD w.r.t. without KD) with *smooth regularization* and *class discriminability* under 270 pairs of "(teacher, student, temperature)". Details are in Appendix C.4. Fig. 3 plots the scatters, where dots show pairs whose improvement is higher than 2%. Clearly, teachers with a larger *derived variance* tend to guide better. *These observations show the rationality of the proposed KD decomposition and imply that enhancing derived variance is beneficial.*

**Larger teachers provide a larger target logit or less varied wrong logits.** We first compare the logit distributions provided by larger and smaller teachers on the training set. Fig. 4 plots the histograms (200 bins) of the target logit (i.e., $\mathbf{f}_y$) and the standard deviation of wrong logits (i.e., $\sigma(\mathbf{g})$). The left and right, respectively, show the results on CIFAR-100 and CIFAR-10. Clearly, the first column shows that ResNet110 tends to generate a larger target logit (i.e., $\mathbb{E}_{\mathbf{x}}[\mathbf{f}_y] \approx 15.0$) than ResNet14 (i.e., $\mathbb{E}_{\mathbf{x}}[\mathbf{f}_y] \approx 10.0$). On CIFAR-10, the smallest $\mathbf{f}_y$ given by WRN28-8 is larger than WRN28-1. Furthermore, WRN28-8 gives smaller variance (the fourth column), i.e., smaller *inherent variance*. *These reveal specific manifestations of complex networks' over-confidence.*

**Larger and smaller teachers have similar inherent variance while different derived variance under traditional temperature scaling.** We use $\tau = 1.0$ to soften the *complete logits* and *only the wrong class logits*, respectively, and then show the mean and standard deviation of *derived variance* and *inherent variance* across training samples in Fig. 5. Although the larger models' *inherent variance* is smaller, the difference between RNX29-4-4d and RNX29-64-4d is only up to $1.3\times$. However, the *derived variance* differs a lot, where the smaller teacher's variance is approximately $9.7\times$ as the larger teacher's. These observations imply that *traditional temperature scaling could seriously decrease the derived variance of larger teachers though they have appreciable inherent variance.*

**Complex teachers know approximately the same as smaller teachers on relative class affinities.** Only if the relative magnitudes of the wrong class probabilities are correct, it is valid to enlarge the discrimination between them. Otherwise, if a teacher himself misunderstands the knowledge's principles, it will be counterproductive to reinforce this knowledge to students. Given two teachers $T_1$ and $T_2$, we first calculate the *Spearman correlation* between $\mathbf{p}^{T_1}$ and $\mathbf{p}^{T_2}$. Second, the set overlap

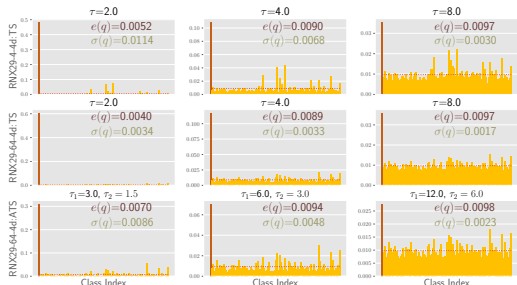

Figure 8: Probability vector visualization of a randomly selected training sample from CIFAR-100. The target class is $y = 1$. The bottom row shows applying ATS to the larger teacher.

Figure 9: Distillation results via TS (solid curves) and ATS (dashed curves) on CIFAR-100. The x-axis of each figure shows teachers with various capacities.

Table 2: Comparisons with SOTA methods on CIFAR-100. ResNet110, WRN28-8, and RNX29-64-4d are teachers. VGG8, SFV1, and MV2 are students. The area in gray shows the results of the ensemble. "KD+ATS" and "KD+ATS+Ens" are our methods.

| Teacher | ResNet110 (74.09) | | | WRN28-8 (79.73) | | | RNX29-64-4d (79.91) | | | Avg |
|---|---|---|---|---|---|---|---|---|---|---|
| Student | VGG8 | SFV1 | MV2 | VGG8 | SFV1 | MV2 | VGG8 | SFV1 | MV2 | |
| NoKD | 69.92 | 70.04 | 64.75 | 69.92 | 70.04 | 64.75 | 69.92 | 70.04 | 64.75 | 68.24 |
| ST-KD | 72.30 | 73.22 | 66.56 | 71.85 | 72.85 | **66.52** | 71.61 | 72.18 | 65.82 | 70.32 |
| KD | 71.35 | 71.86 | 65.49 | 70.46 | 70.87 | 64.97 | 71.13 | 71.80 | 64.99 | 69.21 |
| ESKD | 71.88 | 72.02 | 65.92 | 71.13 | 71.32 | 65.09 | 71.09 | 71.27 | 64.83 | 69.39 |
| TAKD | **72.71** | 72.86 | 66.98 | 71.20 | 71.62 | 65.11 | 71.46 | 71.44 | 65.36 | 69.86 |
| SCKD | 70.38 | 70.61 | 64.59 | 70.83 | 70.52 | 65.19 | 70.33 | 70.92 | 64.86 | 68.69 |
| KD+ATS | 72.31 | **73.44** | **67.18** | **72.72** | **73.58** | 66.47 | **72.93** | **73.03** | **66.80** | **70.94** |
| Ens | 72.77 | 73.61 | 67.76 | 72.77 | 73.61 | 67.76 | 72.77 | 73.61 | 67.76 | 71.38 |
| ResKD | 73.89 | **76.03** | 69.00 | 73.84 | **75.14** | 67.69 | 74.64 | 75.43 | 68.10 | 72.64 |
| KD+ATS+Ens | **74.86** | 75.05 | **69.50** | **74.60** | 75.04 | **68.79** | **75.34** | **75.47** | **69.82** | **73.16** |

between the top-5 predictions is calculated, i.e., $|\mathcal{C}^{T_1} \cap \mathcal{C}^{T_2}|/|\mathcal{C}^{T_1} \cup \mathcal{C}^{T_2}|$. $\mathcal{C}$ denotes the set of top-5 predicted classes. Third, we calculate Kendall's $\tau$ between $\mathbf{p}^{T_1}$ and $\mathbf{p}^{T_2}$, which directly shows the rank correlation of two teachers. These metrics only depend on classes' relative magnitudes. The results are in Fig. 6. Excitingly, these metrics among teachers with different capacities do not vary a lot, and the Spearman correlations are almost all larger than $0.85$. According to the interpretation of Kendall's $\tau$ [8, 53], if the smaller teacher predicts that class $i$ is more related to the target class than that of class $j$, then the larger teacher has a probability of $(0.75 + 1)/2 = 0.875$ to give the same relative affinity. As a comparison, we also calculate the distillation distance defined in [15], which utilizes L1 distance and depends on the absolute magnitudes of probabilities. Using this metric, the distance increases quickly as the capacity gap increases. These observations demonstrate that *teachers know approximately the same about relative class affinities while their absolute values differ significantly under traditional temperature scaling*.

**The proposed ATS could enlarge the derived variance of larger teachers.** The above analysis verifies that the over-confident teachers experience lower *derived variance* under traditional temperature scaling although they grasp the relative class affinities well. For an intuitive visualization, we plot the probabilities for a randomly sampled instance whose correct class is $y = 1$ from CIFAR-100. The top two rows in Fig. 8 show the results of RNX29-4-4d and RNX29-64-4d under traditional temperature scaling (TS), where the latter really experiences a smaller *derived variance*. Using ATS could enhance the *derived variance* as shown at the bottom row (the bars are more jagged). Then, we study the change of *derived average* (DA) and *derived variance* (DV) as temperature increases. Given a $\tau$, we obtain the DA and DV for all training samples via softmax and then calculate the average. For ATS, we use $\tau_1 = 1.25\tau$ and $\tau_2 = 0.75\tau$. The curves are shown in Fig. 7. According to Proposition 4.3, $e(\mathbf{q})$ increases as $\tau$ increases, which enhances the efficacy of *smooth regularization*.

Table 3: Comparisons with SOTA methods on TinyImageNet, CUB, and Stanford Dogs. WRN50-2 and RNX101-32-8d are teachers. AlexNet, SFV2, and MV2 are students.

| | TinyImageNet | | | CUB | | | Stanford Dogs | | | Avg |
|---|---|---|---|---|---|---|---|---|---|---|
| Teacher | WRN50-2 (66.28) | | | RNX101-32-8d (79.50) | | | RNX101-32-8d (73.98) | | | |
| Student | ANet | SFV2 | MV2 | ANet | SFV2 | MV2 | ANet | SFV2 | MV2 | |
| NoKD | 34.62 | 45.79 | 52.03 | 55.66 | 71.24 | 74.49 | 50.20 | 68.72 | 68.67 | 57.94 |
| ST-KD | 36.16 | 49.59 | 52.93 | 56.39 | 72.15 | 76.80 | 51.95 | 69.92 | 72.06 | 59.77 |
| KD | 35.83 | 48.48 | 52.33 | 55.10 | 71.89 | 76.45 | 50.22 | 68.48 | 71.25 | 58.89 |
| ESKD | 34.97 | 48.34 | 52.15 | 55.64 | 72.15 | 76.87 | 50.39 | 69.02 | 71.56 | 59.01 |
| TAKD | 36.20 | 48.71 | 52.44 | 54.82 | 71.53 | 76.25 | 50.36 | 68.94 | 70.61 | 58.87 |
| SCKD | 36.16 | 48.76 | 51.83 | 56.78 | 71.99 | 75.13 | 51.78 | 68.80 | 70.13 | 59.04 |
| KD+ATS | **37.42** | **50.03** | **54.11** | **58.32** | **73.15** | **77.83** | **52.96** | **70.92** | **73.16** | **60.88** |
| Ens | 39.37 | 50.69 | 56.40 | 59.84 | 74.43 | 77.47 | 54.04 | 71.65 | 72.53 | 61.82 |
| ResKD | 38.66 | 51.93 | 57.32 | **62.60** | 75.29 | 76.27 | 54.68 | 70.73 | 72.85 | 62.26 |
| KD+ATS+Ens | **40.42** | **52.14** | **58.47** | 62.00 | **76.26** | **78.97** | **55.69** | **73.22** | **74.67** | **63.54** |

Notably, this term changes nearly the same between teachers with various capacities. However, the *derived variance* $v(\mathbf{q})$ differs a lot. Empirically, $v(\mathbf{q})$ first increases and then decreases, and the maximal of the larger teacher's DV is smaller, which verifies the Corollary 4.7. Because *derived variance* corresponds to the efficacy of *class discriminability*, this shows why larger teachers can't teach well. Using the proposed ATS could enhance the *derived variance*, which equivalently improves the efficacy of *class discriminability* in KD. We conclude that *traditional temperature scaling leads to distillation labels with less discriminative information among wrong class probabilities; our proposed ATS could enhance the discrimination among wrong classes and benefit the distillation process*.

## 5.2 Performances

**ATS makes larger teachers teach well again.** Previous studies find that more accurate teachers can't necessarily teach well [6, 31]. As shown in Fig. 9, although we tune temperatures in $\{1.0, 2.0, 4.0, 8.0, 12.0, 16.0\}$, larger teachers still teach worse under traditional temperature scaling (the solid curves). However, using ATS (the dashed curves) could make larger teachers teach well or better again. The details are in Appendix C.4.

**ATS surpasses previous methods with advanced techniques.** We compare with SOTA methods and list the results on CIFAR-100, TinyImageNet, CUB, and Dogs in Tab. 2 and Tab. 3. The sota compared methods include ESKD [6], TAKD [31], SCKD [51], and ResKD [10, 24]. NoKD trains students without the teacher's supervision. ST-KD trains students under the guidance of a smaller teacher. KD trains students under the guidance of the larger teacher. More details of compared methods are in Appendix C.3. The last column of these tables shows the average performance of corresponding rows. The larger teacher could slightly improve the students' performances via traditional KD, i.e., $68.24\% \rightarrow 69.21\%$ and $57.94 \rightarrow 58.89\%$. However, using a smaller teacher (the row of "ST-KD") could obtain about $2\%$ improvement on average, i.e., $68.24\% \rightarrow 70.32\%$ and $57.94\% \rightarrow 59.77\%$. This again verifies that larger teachers really teach worse on various datasets. Taking advantage of the *early-stopped teacher* (ESKD), the *teacher assistant* (TAKD), and the *student customized teacher* (SCKD) only improves the larger teacher slightly, e.g., $69.21\% \rightarrow 69.86\%$ and $58.89\% \rightarrow 59.04\%$. These results do not even surpass the small teacher. ResKD improves the students' performances a lot via introducing the *residual student* and taking the two students' ensemble, which surpasses the ensemble performances of two separately trained students under different initializations (the "Ens" row). For a fair comparison, we also test the performances of our methods via repeating the "KD+ATS" two times and making predictions via the ensemble. The results are almost $1\%$ higher than ResKD. Results on CIFAR-10 and speech data are in Appendix D.

**Ablation Studies** One might argue that the validity of ATS is due to the tuning from a larger hyper-parameter space. We vary $\tau_1$ from 1.0 to 6.0, $\tau_2$ from 1.0 to 5.0, and record the performances in Fig. 10. Obviously, setting $\tau_1 > \tau_2$ could be better, and especially, we recommend the setting of $\tau_2 \in [\tau_1 - 2, \tau_1 - 1]$. Although we introduce one more hyper-parameter, ATS is simple to implement and surpasses SOTA methods that take advanced techniques. We achieve the goal of only utilizing simple operations to make larger teachers teach well again. Then, we study the trade off of the KD

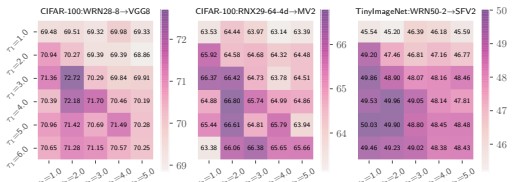

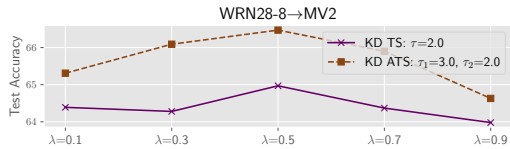

Figure 10: Ablation studies on asymmetric temperatures on CIFAR-100 and TinyImageNet ($\tau_1, \tau_2$ in Eq. 5).

Figure 11: Ablation studies on the weighting of KD loss and CE loss on CIFAR-100 ($\lambda$ in Eq. 1).

loss and CE loss under different $\lambda$. The results of "WRN28-8 → MV2" on CIFAR-100 (Fig. 11) verifies that ATS could improve the performances under various $\lambda$. We also compare ATS with other types of KD under various scenes, and the results are in Appendix D.3.

## 6 Conclusion

We study the miraculous phenomenon in KD that a more accurate model doesn't necessarily teach better. The proposed KD decomposition attributes the success of a better teacher to three factors, including *correct guidance*, *smooth regularization*, and *class discriminability*. Through theoretical analysis, over-confident teachers could not release their potential abilities of the *class discriminability* under traditional temperature scaling. As a simple yet effective solution, we propose *Asymmetric Temperature Scaling (ATS)* to enhance the *derived variance* of larger teachers, making their distillation labels more discriminative when teaching students. Extensive experimental results verify the superiorities of our proposed methods.

## 7 Broader Impact

We focus on the variance of wrong class probabilities to analyze why larger teachers cannot teach well and hope that our research could bring a new perspective to the KD field. Our work has no potential negative societal impacts.

## Acknowledgements

This work is partially supported by National Natural Science Foundation of China (Grant No. 61921006, 41901270), and Natural Science Foundation of Jiangsu Province (Grant No. BK20190296). Thanks to Huawei Noah's Ark Lab NetMIND Research Team. We gratefully acknowledge the support of Mindspore used for this reasearch. Professor De-Chuan Zhan is the corresponding author. A

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
