## A  More Discussions About KD Decomposition (Eq. 2)

We decompose KD into three terms as shown in Eq. 2, i.e., the *correct guidance*, the *smooth regularization*, and the *class discriminability*. We take a reverse thinking that *how can we construct these three terms if without KD?* First, to ensure the learning process of the student is correct, the target class's label should be the highest. This is simple because we have one-hot labels of training samples. That is, the cross-entropy loss could guarantee the student learns correctly. Second, how can we introduce *smooth regularization*? This is also simple because we could utilize label smoothing (LS) as regularizations. However, using cross-entropy and LS is still not as effective as KD. Hence, the *class discriminability* also matters a lot. This term is not easy to construct because we do not have the prior class similarities or the instance-level class similarities. Fortunately, a teacher may contain the information of class similarities in KD. We could find that the *class discriminability* may matter more through the above analysis. Hence, we focus on its efficacy when analyzing why larger models can't teach well.

Furthermore, the *correct guidance* is closely related to the second term, i.e., *smooth regularization*, as $e(\mathbf{q}) = \frac{1}{C-1}(1 - \mathbf{p}_y)$. That is, the strength of *correct guidance* is negatively correlated with the strength of *smooth regularization*. As shown in Fig. 7, after applying ATS, the smooth regularization curves in these three subfigures are nearly the same, which indicates that the *correct guidance* term is slightly affected when utilizing ATS.

## B  Detailed Proofs of Lemma 4.1, Proposition 4.3 and Proposition 4.4

We first present some basic results about softmax operations. The softmax operation is $\mathbf{p}_c = \frac{\exp(\mathbf{f}_c/\tau)}{Z}$, where $Z = \sum_{j=1}^{C} \exp(\mathbf{f_j}/\tau)$. We then present some basic derivatives:

$$\frac{\partial Z}{\partial \tau} = -\sum_{c=1}^{C} \frac{\mathbf{f}_c}{\tau^2} \exp(\mathbf{f}_c/\tau), \quad \frac{\partial \mathbf{p}_c}{\partial \tau} = \frac{\mathbf{p}_c}{\tau^2}\left(\sum_{j=1}^{C} \mathbf{p}_j \mathbf{f}_j - \mathbf{f}_c\right) \tag{6}$$

**Lemma B.1** (Variance of Softened Probabilities (Lemma 4.1 in the Body))**.** *Given a logit vector* $\mathbf{f} \in \mathbb{R}^C$ *and the softened probability vector via softmax function, i.e.,* $\mathbf{p} = SF(\mathbf{f}; \tau), \tau \in (0, \infty)$, $v(\mathbf{p})$ *monotonically decreases as* $\tau$ *increases.*

*Proof.* Obviously, $e(\mathbf{p}) = \frac{1}{C}$. Then, we have:

$$v(\mathbf{p}) = \frac{1}{C} \sum_{c=1}^{C} (\mathbf{p}_c - e(\mathbf{p}))^2 = \frac{1}{C} \sum_{c=1}^{C} \mathbf{p}_c^2 - \frac{1}{C^2}. \tag{7}$$

We take the derivative of $v(\mathbf{p})$ w.r.t. $\tau$, and obtain:

$$\begin{aligned}
\frac{\partial v(\mathbf{p})}{\partial \tau} &= \frac{2}{C} \sum_{c=1}^{C} \mathbf{p}_c \frac{\partial \mathbf{p}_c}{\partial \tau} = \frac{2}{C\tau^2} \sum_{c=1}^{C} \mathbf{p}_c^2 \left(\sum_{j=1}^{C} \mathbf{p}_j \mathbf{f}_j - \mathbf{f}_c\right) \\
&= \frac{2}{C\tau^2}\left(\left(\sum_{c=1}^{C} \mathbf{p}_c^2\right)\left(\sum_{j=1}^{C} \mathbf{p}_j \mathbf{f}_j\right) - \sum_{c=1}^{C} \mathbf{p}_c^2 \mathbf{f}_c\right).
\end{aligned} \tag{8}$$

Substituting $\mathbf{f}_c = \tau \log \mathbf{p}_c + \tau \log Z$, we get:

$$\frac{\partial v(\mathbf{p})}{\partial \tau} = \frac{2}{C\tau}\left(\left(\sum_{c=1}^{C} \mathbf{p}_c^2\right)\left(\sum_{j=1}^{C} \mathbf{p}_j \log \mathbf{p}_j\right) - \sum_{c=1}^{C} \mathbf{p}_c^2 \log \mathbf{p}_c\right). \tag{9}$$

Then, we define $\hat{\mathbf{p}}_c = \mathbf{p}_c^2 / \sum_{c=1}^{C} \mathbf{p}_c^2$, and we could derive the following equation:

$$\frac{\partial v(\mathbf{p})}{\partial \tau} = \frac{2\left(\sum_{c=1}^{C} \mathbf{p}_c^2\right)}{C\tau}\left(\sum_{c=1}^{C} \mathbf{p}_c \log \mathbf{p}_c - \sum_{c=1}^{C} \hat{\mathbf{p}}_c \log \mathbf{p}_c\right). \tag{10}$$

We then calculate the KL divergence of $\hat{\mathbf{p}}$ and $\mathbf{p}$:

$$KL(\hat{\mathbf{p}}\|\mathbf{p}) = \sum_{c=1}^{C} \hat{\mathbf{p}}_c \log \frac{\hat{\mathbf{p}}_c}{\mathbf{p}_c} = \sum_{c=1}^{C} \hat{\mathbf{p}}_c \log \frac{\mathbf{p}_c}{\sum_j \mathbf{p}_j^2} = \sum_{c=1}^{C} \hat{\mathbf{p}}_c \log \mathbf{p}_c - \sum_{c=1}^{C} \hat{\mathbf{p}}_c \log\left(\sum_{j=1}^{C} \mathbf{p}_j^2\right). \tag{11}$$

Due the non-negativity of KL divergence, we have:

$$\sum_{c=1}^{C} \hat{\mathbf{p}}_c \log \mathbf{p}_c \geq \sum_{c=1}^{C} \hat{\mathbf{p}}_c \log \left( \sum_{j=1}^{C} \mathbf{p}_j^2 \right) = \log \left( \sum_{c=1}^{C} \mathbf{p}_j^2 \right). \tag{12}$$

Hence, we have:

$$\sum_{c=1}^{C} \mathbf{p}_c \log \mathbf{p}_c - \sum_{c=1}^{C} \hat{\mathbf{p}}_c \log \mathbf{p}_c \leq \sum_{c=1}^{C} \mathbf{p}_c \log \mathbf{p}_c - \log \sum_{c=1}^{C} \mathbf{p}_c^2 \leq 0, \tag{13}$$

where the first inequality is according to Eq. 12 and the last inequality is according to Jensen's inequality, This proves that $\frac{\partial v(\mathbf{p})}{\partial \tau} \leq 0$. □

**Proposition B.2** ((Proposition 4.3 in the Body)). *Under the Assumption 4.2, $\mathbf{p}_y$ monotonically decreases as $\tau$ increases, and $e(\mathbf{q})$ monotonically increases as $\tau$ increases. As $\tau \to \infty$, $e(\mathbf{q}) \to 1/C$.*

*Proof.* We take the derivative of $\mathbf{p}_y$ w.r.t. $\tau$, and obtain:

$$\frac{\partial \mathbf{p}_y}{\partial \tau} = \frac{\mathbf{p}_y}{\tau^2} \left( \sum_{c=1}^{C} \mathbf{p}_c \mathbf{f}_c - \mathbf{f}_y \right) \leq 0, \tag{14}$$

which shows the monotonicity of $\mathbf{p}_y$ w.r.t. $\tau$. Then the later is obvious because $e(\mathbf{q}) = \frac{1}{C-1} \sum_{c \neq y} \mathbf{p}_c = \frac{1}{C-1}(1 - \mathbf{p}_y)$. Finally, according to the properties of limitation, $\lim_{\tau \to \infty} \mathbf{p}_y = \lim_{\tau \to \infty} \frac{\exp(\mathbf{f}_y/\tau)}{\sum_{j=1}^{C} \exp(\mathbf{f}_j/\tau)} = 1/C$. □

**Proposition B.3** (Derived Variance vs. Inherent Variance (Proposition 4.4 in the Body)). *The derived variance is determined by the square of derived average and the inherent variance via the following equation:*

$$\underbrace{v(\mathbf{q})}_{DV} = (C-1)^2 \underbrace{e^2(\mathbf{q})}_{DA^2} \underbrace{v(\tilde{\mathbf{q}})}_{IV}. \tag{15}$$

*Proof.* With the property in Eq. 3, we have: $\tilde{\mathbf{q}}_{c'} = \frac{\exp(\mathbf{g}_{c'}/\tau)}{\sum_j \exp(\mathbf{g}_j/\tau)} = \frac{\mathbf{p}_c}{\sum_{j \neq y} \mathbf{p}_j} = \frac{\mathbf{p}_c}{1 - \mathbf{p}_y}$. $c'$ is the corresponding index of $\mathbf{p}_c$ in $\mathbf{q}$ after removing the correct class probability $\mathbf{p}_y$, i.e., $c' = c$ when $c < y$, and $c' = c - 1$ when $c > y$.

$$\begin{aligned} v(\mathbf{q}) &= \frac{1}{C-1} \sum_{c \neq y} \mathbf{p}_c^2 - e^2(\mathbf{q}) = \frac{1}{C-1} \sum_{c \neq y} \mathbf{p}_c^2 - \frac{(1-\mathbf{p}_y)^2}{(C-1)^2} \\ &= (1-\mathbf{p}_y)^2 \left( \frac{1}{C-1} \sum_{c \neq y} \left( \frac{\mathbf{p}_c}{1-\mathbf{p}_y} \right)^2 - \frac{1}{(C-1)^2} \right) \\ &= (1-\mathbf{p}_y)^2 \left( \frac{1}{C-1} \sum_{c'} \tilde{\mathbf{q}}_{c'}^2 - \frac{1}{(C-1)^2} \right) \\ &= (1-\mathbf{p}_y)^2 v(\tilde{\mathbf{q}}) = (C-1)^2 e^2(\mathbf{q}) v(\tilde{\mathbf{q}}). \end{aligned} \tag{16}$$

□

## C  Details of Datasets, Networks and Training

### C.1  Dataset Details

The datasets used in our experiments are CIFAR-10/CIFAR-100 (C10/C100) [21], TinyImageNet (TIN) [43], CUB [47], Stanford Dogs (Dogs) [19], and Google Speech Commands (GSC) [48].

CIFAR-10/CIFAR-100 are image classification datasets, and each contains 50K training and 10K test samples of size $32 \times 32$. TinyImageNet contains 200 classes, 100K training samples, and 10K

Table 4: Statistics of datasets. **Task**: Image Classification (IC), Fine-Grained Recognition (FGR), Keyword Spotting (KWS). **Size**: the input size of a single sample, we omit the dimension of channel. **Networks**: ResNet (RES), WideResNet (WRN), ResNeXt (RNX), ShuffleNetV1/2 (SFV1/2), MobileNetV2 (MV2), AlexNet (ANet).

| Dataset | Task | Size | C | Num.Tr | Num.Te | Teacher Networks | Student Networks |
|---------|------|------|---|--------|--------|------------------|------------------|
| C10 | IC | (32, 32) | 10 | 50K | 10K | RES, WRN, RNX | VGG8, SFV1, MV2 |
| C100 | IC | (32, 32) | 100 | 50K | 10K | RES, WRN, RNX | VGG8, SFV1, MV2 |
| TIN | IC | (64, 64) | 200 | 100K | 10K | WRN50-2 | ANet, SFV2, MV2 |
| CUB | FGR | (224, 224) | 200 | 6K | 5.8K | RNX101-32-8d | ANet, SFV2, MV2 |
| Dogs | FGR | (224, 224) | 120 | 12K | 8.6K | RNX101-32-8d | ANet, SFV2, MV2 |
| GSC | KWS | (101, 40) | 35 | 106K | 11K | RES | DSCNN |

Table 5: Details of networks and their performances on training and test set (Accs). "RNX" is abbreviated as "R" to save space. Student networks are shaded while teachers are not.

| Network | N.P | Accs | Network | N.P | Accs | Network | N.P | Accs |
|---------|-----|------|---------|-----|------|---------|-----|------|
| **CIFAR-10 Teachers** | | | | | | | | |
| ResNet | | | WideResNet | | | ResNeXt | | |
| RES14 | 0.18M | 98.4, 91.5 | WRN28-1 | 0.37M | 99.7, 92.8 | R29-4-4d | 1.2M | 100, 93.9 |
| RES20 | 0.27M | 99.5, 92.3 | WRN28-2 | 1.5M | 100, 94.9 | R29-8-4d | 1.7M | 100, 94.7 |
| RES32 | 0.47M | 99.9, 93.5 | WRN28-3 | 3.3M | 100, 95.3 | R29-16-4d | 2.7M | 100, 95.2 |
| RES44 | 0.66M | 99.9, 93.8 | WRN28-4 | 5.8M | 100, 95.6 | R29-24-4d | 3.8M | 100, 95.3 |
| RES56 | 0.86M | 100, 93.9 | WRN28-6 | 13M | 100, 95.9 | R29-32-4d | 4.8M | 100, 95.6 |
| RES110 | 1.7M | 100, 94.3 | WRN28-8 | 23M | 100, 96.1 | R29-64-4d | 8.9M | 100, 95.8 |
| **CIFAR-10 Students** | | | | | | | | |
| VGG8 | 3.9M | 99.9, 91.7 | SFV1 | 0.86M | 99.9, 92.1 | MV2 | 0.7M | 92.6, 88.4 |
| **CIFAR-100 Teachers** | | | | | | | | |
| ResNet | | | WideResNet | | | ResNeXt | | |
| RES14 | 0.18M | 81.1, 67.0 | WRN28-1 | 0.38M | 91.8, 70.0 | R29-4-4d | 1.3M | 99.8, 75.1 |
| RES20 | 0.28M | 88.1, 69.1 | WRN28-2 | 1.5M | 99.8, 74.2 | R29-8-4d | 1.8M | 99.9, 76.0 |
| RES32 | 0.47M | 94.3, 71.1 | WRN28-3 | 3.3M | 100, 76.7 | R29-16-4d | 2.8M | 100, 77.9 |
| RES44 | 0.67M | 97.3, 72.2 | WRN28-4 | 5.9M | 100, 77.9 | R29-24-4d | 3.8M | 100, 79.1 |
| RES56 | 0.86M | 98.7, 72.9 | WRN28-6 | 13M | 100, 79.1 | R29-32-4d | 4.9M | 100, 79.3 |
| RES110 | 1.7M | 99.8, 74.1 | WRN28-8 | 23M | 100, 79.7 | R29-64-4d | 8.9M | 100, 79.9 |
| **CIFAR-100 Students** | | | | | | | | |
| VGG8 | 4.0M | 99.8, 69.9 | SFV1 | 0.95M | 99.9, 70.0 | MV2 | 0.81M | 83.7, 64.8 |
| **TinyImageNet Teachers** | | | | | | | | |
| - | - | - | WRN50-2 | 67M | 100, 66.3 | - | - | - |
| **TinyImageNet Students** | | | | | | | | |
| ANet | 2.7M | 98.5, 34.6 | SFV2 | 1.5M | 94.3, 45.8 | MV2 | 2.5M | 85.9, 52.0 |
| **CUB Teachers** | | | | | | | | |
| - | - | - | - | - | - | R101-32-8d | 87M | 98.1, 79.5 |
| **CUB Students** | | | | | | | | |
| ANet | 2.7M | 92.8, 55.7 | SFV2 | 1.5M | 93.7, 71.2 | MV2 | 2.5M | 95.8, 74.5 |
| **Stanford Dogs Teachers** | | | | | | | | |
| - | - | - | - | - | - | R101-32-8d | 87M | 97.8, 74.0 |
| **Stanford Dogs Students** | | | | | | | | |
| ANet | 2.6M | 88.0, 50.2 | SFV2 | 1.4M | 92.6, 68.7 | MV2 | 2.4M | 94.5, 68.7 |
| **Google Speech Commands Teachers** | | | | | | | | |
| RES | 1.9M | 98.8, 99.2 | - | - | - | - | - | - |
| **Google Speech Commands Students** | | | | | | | | |
| DSCNN | 59.8K | 93.2, 94.5 | - | - | - | - | - | - |

test samples of size $64 \times 64$. CUB and Stanford Dogs are fine-grained recognition datasets, which correspondingly contain 200 and 120 classes. Google Speech Commands is a benchmark for keyword spotting [60, 41], which usually needs efficient model deployment. It aims to identify whether a 1s-long speech recording is a word, silence, or unknown. There are total 35 classes (35 words) in this task. We extract 40 MFCC features for each 30ms window frame with a stride of 10ms. We also follow the settings in Google Speech Commands (GSC) [48]: performing random time-shift of $Y \in [-100, 100]$ milliseconds and adding 0.1 volume background noise with a probability of 0.8. The details are listed in Tab. 4. $C$ denotes the number of classes. "Num.Tr" and "Num.Te" denote the number of training and test samples.

## C.2 Network Details

For teacher networks, we use different versions of ResNet [11], WideResNet [56], ResNeXt [49]. For student networks, we use VGG [39], ShuffleNetV1/V2 [59, 29], AlexNet [22], MobileNetV2 [37], and DSCNN [60].

- **ResNet (RES)**: We utilize ResNet for CIFAR-10/100 and Google Speech Commands as teachers. For CIFAR-10/100, we use the original ResNet versions with "6n+2" layers in [11]. These ResNets only use the "basic block" for CIFAR data. Specifically, we vary the number of layers in $\{14, 20, 32, 44, 56, 110\}$. For Google Speech Commands, we utilize the ResNet version proposed in [41]. We modify the number of layer as 15 and the channel as 128.
- **WideResNet (WRN)**: We utilize WideResNet for CIFAR-10/100 and TinyImageNet as teachers. For CIFAR-10/100, we use the WideResNet proposed in [56]. We keep the depth as 28 and vary the widen factors in $\{1, 2, 3, 4, 6, 8\}$. For TinyImageNet, we use the WideResNet50-2 provided in PyTorch [2].
- **ResNeXt (RNX)**: We utilize ResNeXt for CIFAR-10/100, CUB and Stanford Dogs as teachers. For CIFAR10-/100, we utilize the original ResNeXt versions for CIFAR data as proposed in [49]. We take 29 layers and set the base width as 4, varying the number of groups in $\{4, 8, 16, 24, 32, 64\}$. For CUB and Stanford Dogs, we use the ResNeXt101-32-8d provided in PyTorch.
- **VGG8**: We utilize VGG8 [39] for CIFAR-10/100 as the student. Although it contains more parameters, the plain architecture makes it weaker.
- **ShuffleNetV1/2 (SFV1/2)**: We utilize ShuffleNet for CIFAR-10/100, TinyImageNet, CUB, and Dogs as students. For CIFAR-10/100, we use ShuffleNetV1 [59] provided in RepDistiller repository [3]. For other datasets, we use ShuffleNetV2 [29] provided in PyTorch.
- **MobileNetV2 (MV2)**: We utilize MobileNet for CIFAR-10/100, TinyImageNet, CUB, and Dogs as students. For CIFAR-10/100, we use MobileNetV2 [37] in RepDistiller repository. For other datasets, we use ShuffleNetV2 [37] provided in PyTorch.
- **AlexNet (ANet)**: We utilize AlexNet provided in PyTorch for TinyImageNet, CUB, and Dogs as students.
- **DSCNN**: We use DSCNN [60] for Google Speech Commands as the student. It utilizes depth-separable convolution to accelerate training and inference. We set the basic number of channels as 96.

Additionally, for CUB and Stanford Dogs, we use the corresponding pre-trained models provided in PyTorch as initialization. The number of parameters ("N.P") and the training and test accuracies are in Tab. 5.

## C.3 Training Details

**Train Teachers** We first train teacher networks on corresponding datasets and then save the checkpoints of them.

**Train Students** We train students via the loss function in Eq. 1. We set $\lambda = 0.5$ by default and tune $\tau \in \{1.0, 2.0, 4.0, 8.0, 12.0, 16.0\}$. Additionally, we vary the student's $\tau$ in two settings: (1) the same

---

[2] `https://pytorch.org/vision/stable/models.html`
[3] `https://github.com/HobbitLong/RepDistiller/tree/master/models`

Table 6: Comparisons with SOTA methods on CIFAR-10. ResNet110, WRN28-8, and RNX29-64-4d are used as teachers. VGG8, SFV1, and MV2 are students.

| Teacher | ResNet110 (94.3) | | | WRN28-8 (96.1) | | | RNX29-64-4d (95.8) | | | Avg |
|---|---|---|---|---|---|---|---|---|---|---|
| Student | VGG8 | SFV1 | MV2 | VGG8 | SFV1 | MV2 | VGG8 | SFV1 | MV2 | |
| NoKD | 91.68 | 92.11 | 88.36 | 91.68 | 92.11 | 88.36 | 91.68 | 92.11 | 88.36 | 90.72 |
| ST-KD | 92.47 | 93.20 | **88.83** | 92.48 | 93.19 | **88.72** | 92.56 | 93.20 | 88.70 | 91.48 |
| KD | 92.29 | 92.80 | 88.60 | 92.35 | 93.07 | 88.54 | 92.44 | 93.18 | 88.37 | 91.29 |
| ESKD | 92.07 | 92.84 | 88.18 | 92.06 | 92.84 | 88.33 | 92.32 | 92.90 | 88.21 | 91.08 |
| TAKD | 92.10 | 93.13 | 88.36 | 92.33 | 92.59 | 88.15 | 92.31 | 93.03 | 88.50 | 91.17 |
| SCKD | 91.75 | 92.79 | 88.70 | 91.66 | 92.57 | 88.50 | 91.91 | 92.59 | 88.30 | 90.97 |
| KD+ATS | **92.88** | 92.99 | 88.63 | **92.84** | **93.20** | 88.43 | **92.73** | **93.27** | **88.96** | **91.55** |

as the teacher's; (2) 1.0. For our proposed KD method, the pair of $(\tau_1, \tau_2)$ in Eq. 5 is searched in $\{(2.0, 1.0), (3.0, 1.0), (3.0, 2.0), (4.0, 2.0), (4.0, 3.0), (5.0, 2.0)\}$. For our method, we keep the student's temperature as 1.0 by default.

**Compared Methods** We then explain the compared methods in Tab. 2 and Tab. 3. For CIFAR-100/CIFAR-10, we use ResNet110, WRN28-8, and RNX29-64-4d as teachers. For TinyImageNet, CUB, and Stanford Dogs, we use WRN50-2 and ResNeXt101-32-8d as teachers. These models are complex networks with a larger depth, or a larger widen factor, or more groups. We have also explored the standard deviation of these KD learning methods. KD performances are relatively stable among several training random seeds with the same hyper-parameter settings. The standard deviations are also nearly the same across different KD methods, which are about $0.1\%$. Hence, we do not report the standard deviations due to space limitation.

- **NoKD**: training the student without distillation.

- **ST-KD**: training the student under the guidance of a smaller teacher. For CIFAR-100, we select the best performance from all smaller networks as shown in Fig. 9. For TinyImageNet, CUB, and Stanford Dogs, we correspondingly use WRN26-2, RNX50-32-4d, RNX50-32-4d as smaller teachers.

- **KD**: training the student under the guidance of the larger teacher.

- **ESKD** [6]: training the student under the guidance of the *early-stopped teacher*. Specifically, we train the larger teachers only for $\{60, 120, 150\}$ epochs using the cosine annealing learning rate and report the best accuracy of the student.

- **TAKD** [31]: training the student under the guidance of the *teacher assistant*. Specifically, we vary the TA in {ResNet20, ResNet44} for ResNet110, {WRN28-2, WRN28-4} for WRN28-8, {RNX29-8-4d, RNX29-8-24d} for RNX29-64-4d, {WRN26-2} for WRN50-2, and {RNX50-32-4d} for RNX101-32-8d.

- **SCKD** [51]: training the student under the *student customized teacher*. We use two KD losses, including the KL divergence of probabilities (Eq. 1) and the L2 feature distance loss as done in [36].

- **Ens**: training two students with different initializations two times and taking their ensemble.

- **ResKD**: training the student first and then training the *residual student* to fit the residual. The ensemble of these two students is used for inference.

- **KD+ATS**: training the student with the larger teacher's guidance via the proposed ATS.

- **KD+ATS+Ens**: repeating the above two times and taking the ensemble results. When repeating "KD+ATS" two times, we use the same hyperparameters and keep the fairness of comparisons.

## C.4 Figure Details

We explain some figures in detail.

Table 7: Comparisons with other KD methods on CIFAR-100. The results of first five columns are cited from [44]. The gray area shows the results of scenes with smaller "capacity gap". ReKD denotes ReviewKD [4].

| Scene | Teacher | Student | KD | CC | NST | CRD | IE-AT | ReKD | Ours |
|---|---|---|---|---|---|---|---|---|---|
| ResNet110 → ResNet20 | 74.31 | 69.06 | 70.67 | 69.48 | 69.53 | 71.14 | 71.34 | 71.37 | **71.57** |
| VGG13 → VGG8 | 74.64 | 70.36 | 72.98 | 70.71 | 71.53 | 73.94 | **74.05** | 73.60 | 73.65 |
| ResNet56 → ResNet20 | 72.34 | 69.06 | 70.66 | 69.63 | 69.60 | **71.16** | 70.87 | 70.95 | 70.99 |
| WRN40-2 → WRN16-2 | 75.61 | 73.26 | 74.92 | 73.56 | 73.68 | 75.48 | 74.80 | **75.67** | 75.03 |

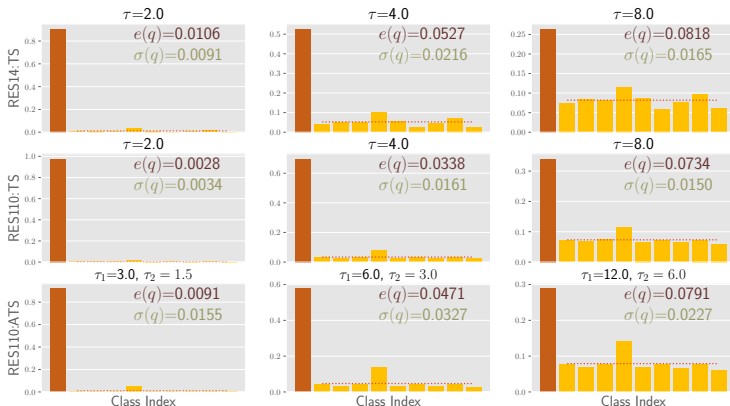

Figure 12: Probability vector visualization of a randomly selected training sample in CIFAR-10. The networks are various ResNet. The target class is $y = 1$. The bottom row shows the efficacy of ATS for a larger teacher.

- Fig. 2: for "ResNet→SFV1", we use ResNet14 as the small teacher ("ST") and ResNet110 as the large teacher ("LT"); for "WRN→MV2", we use WRN28-1 as the small teacher ("ST") and WRN28-8 as the large teacher ("LT").

- Fig. 3: on CIFAR-100, we vary 18 teachers and 3 students as listed in Tab. 5; we also vary temperatures in {1.0, 2.0, 4.0, 8.0, 16.0}; the total pairs of distillation combinations are $18 \times 3 \times 5 = 270$; for each pair, we calculate the teacher's *derived average* and *derived variance* averaged across 50K training samples; the KD improvement ratio is calculated by $(\text{Acc}_{\text{KD}} - \text{Acc}_{\text{NoKD}})/\text{Acc}_{\text{NoKD}}$.

- Fig. 6 and Fig. 5: the former calculates $\mathbf{f}_y$ (i.e., target logit) and $\sigma(\mathbf{g})$ (i.e., standard deviation of wrong logits) across training samples and plot the bins; the latter calculates $\sigma(\mathbf{q})$ (i.e., derived variance) and $\sigma(\tilde{\mathbf{q}})$ (i.e., inherent variance) and only report their mean and standard deviation across training samples.

- Fig. 4: for each pair of two teachers $T_1$ and $T_2$, we obtain their softened labels on all training samples; for each sample, we calculate the four metrics and report the average across training samples.

- Fig. 7: given a temperature $\tau$, we could obtain softened probabilities of all training samples; then we calculate $\sigma(\mathbf{q})$ (i.e., derived variance) and $\sigma(\tilde{\mathbf{q}})$ (i.e., inherent variance) across all training samples and only report the average results.

- Fig. 9: For TS, we tune $\tau \in \{1.0, 2.0, 4.0, 8.0, 12.0, 16.0\}$ and vary the student's $\tau$ in two settings: (1) the same as the teacher's, (2) 1.0; For ATS, the pair of $(\tau_1, \tau_2)$ in Eq. 5 is searched in $\{(2.0, 1.0), (3.0, 1.0), (3.0, 2.0), (4.0, 2.0), (4.0, 3.0), (5.0, 2.0)\}$; through adjusting the hyper-parameters and selecting the best results, we plot the performance curves.

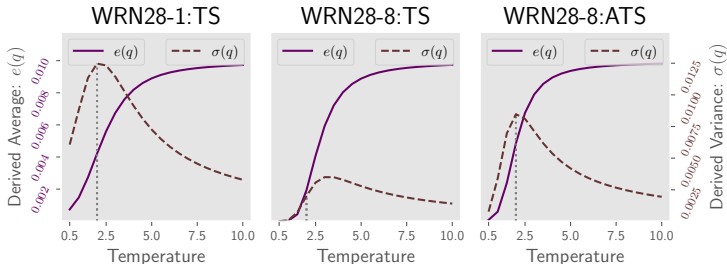

Figure 13: The change of *derived average* ($e(\mathbf{q})$) and *derived variance* ($v(\mathbf{q})$) as $\tau$ increases from 0.1 to 10.0 on CIFAR-100 using various WRN.

Table 8: Comparisons on Google Speech Commands under various types of KD. $+/-$ denotes multiplying $\tau^2$ or not in Eq. 1.

| $(\lambda, +/-)$ | $(0.5, +)$ | $(0.9, +)$ | $(0.5, -)$ | $(0.9, -)$ |
|---|---|---|---|---|
| KD+TS | 95.52 | 95.37 | 94.55 | 94.37 |
| KD+ATS | 96.79 | 97.21 | 96.23 | 95.84 |

# D   More Experimental Studies

## D.1   CIFAR-10 Results

We also compare our proposed methods with SOTA on CIFAR-10, and the results are listed in Tab. 6. Because CIFAR-10 only contains 10 classes and is an slightly simpler benchmark, the performance improvement is not so obvious as other datasets.

## D.2   Speech Data Results

ATS could also perform well on speech data, i.e., the Google Speech Commands benchmark as shown in Tab. 8, where we vary several types of KD loss. Our proposed ATS does not depend on multiplying $\tau^2$ or not in Eq. 1.

## D.3   Comparisons With Other KD Methods on CIFAR-100

We also compare our proposed method with other KD methods, such as [44, 17, 34, 4, 16]. The previous work [44] provides a comprehensive experimental study on many KD methods. For fair comparison, we directly cite their results and only report several scenes of "Larger Teacher $\rightarrow$ Smaller Student", i.e., "ResNet110 ($74.31\%$) $\rightarrow$ ResNet20 ($69.06\%$)" and "VGG13 ($74.64\%$) $\rightarrow$ VGG8 ($70.36\%$)". For IE-KD [16], we use the proposed variant of IE-AT. For IE-KD [16] and ReviewKD [4], we utilize the public code that the authors provide. We also compare with these two methods because they are the recently proposed SOTA KD methods. The results are listed in Tab. 7. Consistently, our method could improve the KD up to $1\%$ and the results are comparable with CRD that utilizes more advanced techniques. We also apply ATS to the cases that students have almost the same capacity as teachers, e.g., "WRN40-2 ($75.61\%$) $\rightarrow$ WRN16-2 ($73.26\%$)" and "ResNet56 ($72.34\%$) $\rightarrow$ ResNet20 ($69.06\%$)". Under these scenes, our method becomes not so effective and the performance improvement over KD seems more like the benefits of hyper-parameter tuning. Hence, our proposed ATS may be more effective when applying a more complex teacher to guide the learning process of a smaller student.

## D.4   More Results for the Observations

We also present additional results for the observations in Fig. 8 and Fig. 7 to verify that the observations and conclusions are not accidental. Similar results are shown in Fig. 12 and Fig. 13.