# OpenReview forum: "Asymmetric Temperature Scaling Makes Larger Networks Teach Well Again"
_NeurIPS.cc/2022/Conference — NeurIPS 2022 Accept_

### Official Review · Reviewer_HVhD · 2022-07-08

**Rating:** 6
**Confidence:** 3
**Soundness:** 3 good
**Presentation:** 3 good
**Contribution:** 3 good

**Summary:**

This paper revisits the phenomenon in knowledge distillation that a teacher model with better performance doesn’t necessarily transfer information better to the student model. By revealing that the scaling temperature value has a different influence on three terms, correct guidance, smooth regularization and class discriminability, the authors propose to set asymmetric temperature value for the distillation loss function, particularly encouraging more derived variance to improve the distillation better. Experimental results on commonly used vision models have been shown to prove the effectiveness of the proposed model.

**Questions:**

1  When repeating the “KD+ATS” two times and making predictions via ensemble, do the tau1 and tau2 keep the same or not?

2 Compared with ResKD, the proposed method generally performs worse when student model is SFV1 and the only better one is 75.47 over 75.43 which is very insignificant. Can the authors provide any explanation for this?


**Limitations:**

Yes

**Strengths And Weaknesses:**

Strength
1 The paper studies a commonly ignored problem in knowledge distillation, with a relatively simple yet effective solution to solve it.

2 The figures to illustrate the relation between different terms and concept are clearly displayed.


Weaknees

1 Some introduction about the compared methods should be put in the main content rather than the supplementary materials. It is difficult to relate the shorname in Table 2 to corresponding methods and tell the significance of the contribution for the proposed method.

2 For marginal improvements, there lacks of standard deviation to tell the significance. For example, the result in Table2 for RNX29, the last two rows under SFC1 column (75.47 vs 75.43)

---

> ### Author Response · Authors · 2022-08-01
> **Responses to Reviewer HVhD**
>
> [Q1] The standard deviation to tell the significance.
>
> [A1] We have indeed explored the standard deviation of KD learning process. KD performances are relatively stable among several training random seeds with the same hyper-parameter settings. The standard deviations are also nearly the same across different KD methods, which are about $0.1\%$. Hence, we do not report the standard deviations due to space limitation. Although the performance of “KD+ATS+Ens” on RNX29-64-4d $\rightarrow$ SFV1 in Table.1 is nearly the same as “ResKD” without any obvious advantage, the results on other columns indeed show performance gains.
>
> [Q2] When repeating the “KD+ATS” two times and making predictions via ensemble, do the tau1 and tau2 keep the same or not?
>
> [A2] We use the same hyperparameters and keep the fairness of comparisons.
>
> [Q3] The effectiveness of ATS when the student network is SFV1.
>
> [A3] Thanks for the reviewer’s observation that we did not notice before. This is not related to the student networks with ShuffleNet architectures (see Table.2 SFV2 columns). Hence, this may be determined by the special properties of training ShuffleNet on CIFAR-100, where “ResKD” may be more applicable. To verify this guess, we study another four pairs of comparisons on CIFAR-100: WRN28-6 $\rightarrow$ SFV1, WRN28-4 $\rightarrow$ SFV1, RNX29-24-4d $\rightarrow$ SFV1, RNX29-32-4d $\rightarrow$ SFV1. For each experiment, we run three times and report the mean and standard deviation of test accuracies. Performances of “ResKD” are 75.32 (0.012), 75.69 (0.009), 75.95 (0.016), 76.23 (0.012). Performances of “KD+ATS+Ens” are 75.05 (0.013), 75.87 (0.014), 75.87 (0.008), 76.18 (0.013). Hence, we find that “ResKD” indeed performs slightly well on CIFAR-100 with SFV1 as the student. The principle behind it is interesting and we will explore it more in future work.

---

### Official Review · Reviewer_iTrh · 2022-07-11

**Rating:** 6
**Confidence:** 5
**Soundness:** 3 good
**Presentation:** 4 excellent
**Contribution:** 3 good

**Summary:**

This paper proposes to use asymmetric temperature scaling to modify the correct class / wrong class logits of teacher networks, to achieve a more efficient distillation. The logic behind ATS is that logits from complex teachers are typically over-confident and less informative, thus by applying high temperature on correct class and low temperature on wrong classes, the logits could become both confident and discriminative. Both theoretical analysis and experimental results are provided to support the effectiveness of the proposed ATS strategy.

**Questions:**

As listed in the strength and weakness part, my question is primarily regarding if the improvement on smooth regularization and class discrimination comes at a cost of reducing correct guidance, as well as some improvement on the experimental part.

**Strengths And Weaknesses:**

Strength:

The idea of ATS is novel. I also like the presentation of this paper. Easy & clear to follow.

Weakness:

Theoretical part is a bit confusing. Reading the decomposition of KD losses, while the proposed ATS improves the latter two terms, wouldn't it hurt the first term, the correct guidance term? How would the trade-off look like?

I also believe that the experimental part could be much improved. The authors did provide a lot of details and analysis on their results already, but I would like to see the comparisons to more SOTA methods on KD, e.g.,

1) https://arxiv.org/pdf/2107.00181.pdf
2) https://arxiv.org/pdf/2104.09044.pdf

---

> ### Author Response · Authors · 2022-08-01
> **Responses to Reviewer iTrh**
>
> [Q1] Wouldn't ATS hurt the first term, the correct guidance term? How would the trade-off look like?
>
> [A1] The correct guidance is closely related to the second term, i.e., smooth regularization, as $\mathbb{E}[\mathbf{q}] = \frac{1}{C-1} (1-\mathbf{p}_y)$. That is, the strength of correct guidance is negatively correlated with the strength of smooth regularization. As shown in Fig.7, after applying ATS, the smooth regularization curves in these three subfigures are nearly the same, which indicates that the correct guidance term is slightly affected. Thanks for the reviewer’s question. We will explain this more clearly in the final version if possible.
>
> [Q2] Comparisons with SOTA KD methods.
>
> [A2] In Table. 6 (supp), we compare ATS with several SOTA KD methods. The conclusion is that ATS could lead to improvements for benchmarks with complex teachers. We have also verified the efficacy of ReviewKD (recommended by the reviewer) on ResNet110 $\rightarrow$ ResNet20. The utilized code is publicly provided by ReviewKD authors. The student’s accuracy is $71.37$, which is lower than ours. We will add more experimental results of these SOTA KD methods in the future version.

---

> ### Comment · Reviewer_iTrh · 2022-08-09
> **After-rebuttal**
>
> Appreciate the authors for the revision. I am increasing my score after-rebuttal.

---

### Official Review · Reviewer_RAjp · 2022-07-11

**Rating:** 6
**Confidence:** 3
**Soundness:** 2 fair
**Presentation:** 2 fair
**Contribution:** 2 fair

**Summary:**

Working in the teacher-student setting, the authors propose and study a possible method of improving the efficacy of the teacher network by manipulating the temperature parameter of its probability density over labels. They are motivated by the practical need to distill the ‘knowledge’ of the teacher into smaller networks that can be run locally on small devices.

**Questions:**

There is a serious lack of clarity in the exposition. For instance, the core technical motivation of the paper seems to be that larger/more complex/more accurate networks ought to be better teachers (than smaller ones performing the same task), but this is not at all an obvious point. Why do the authors have this expectation?

In Section 3, the teacher-student framework within which the authors are working is not carefully set up. The reader is able to infer that the input to the student network is a probability density over labels generated by the teacher network, but this is not explicitly stated, and no further details of the training problem to be solved by the student (or the details of the teacher network) are provided. The authors might consider rewriting this section to be more informative. Once the mathematical framework is established, it will be easier to demonstrate what the problem is with the output of a large teacher network that prevents it from teaching as effectively as a smaller teacher network, which is what the authors try to do in Section 4. It will also make it easier to precisely characterize the relevance of the temperature parameter to this problem.

In Section 4.1, a decomposition of the loss function of the teacher is made. Per my understanding, this decomposition is then used to motivate ATS. Unfortunately, the decomposition itself is ill-defined - the quantity $E_{j\neq y}$ is not defined at all. This makes it difficult to understand and interpret. If the authors could specify the meaning of this quantity, that would be helpful.

The central quantities the authors ultimately work with are $\mathbb{E}(p)$ and $\mathbb{V}(p)$. (These are introduced in Section 4.2.) However, $p$ is a normalized probability density over labels, and not a random variable. Therefore it is not meaningful to talk about its ‘mean’ and ‘variance’. I’d appreciate it if the authors could clarify exactly the meaning of these quantities. There is a similar problem for the quantities $\mathbb{E}(q)$ and $\mathbb{V}(q)$, where $q$ is a vector of probabilities constructed from $p$ by simply removing the entry corresponding to the correct label. For this reason, I am unable to understand the content of Section 4.2 and therefore much of the rest of the paper.

If the authors are able to edit the paper for clarity, addressing the various points I have mentioned, it will make it easier to try to assess their contribution.

Lastly, I wonder how the idea of asymmetric temperatures affects the input-output map of the teacher network? In particular, if the probability of the correct label corresponding to some input example has one associated 'temperature', and all the other labels have another, then the equilibrium distribution of the resulting map between inputs and label probabilities is no longer the Boltzmann distribution. I imagine this has profound consequences for the map between (teacher) inputs and (teacher) labels. I'd be grateful if the authors could please comment on this point.

**Limitations:**

Yes.

**Strengths And Weaknesses:**

This paper treats an interesting and relevant problem, namely, how to increase the information content of the output of the teacher network. They propose a novel method they call asymmetric temperature scaling (ATS) as one potential solution to this problem. However, I think there are two kinds of issues that significantly weaken the work to the point where I really cannot recommend acceptance: (1) the mathematical basis of the proposed method is ill-defined and shaky, and (2) there is a marked lack of clarity in the writing. Core mathematical details on which the analysis of the paper rests are not properly laid out, making it difficult to assess the contribution of the paper. I suspect the issues are structural and cannot be fixed over the course of a review period. While I am happy to be convinced otherwise, for now I am afraid I must recommend that the paper be rejected.

---

> ### Author Response · Authors · 2022-08-01
> **Responses to Reviewer RAjp**
>
> [Q1] Larger networks ought to be better teachers. This is not at all an obvious point.
>
> [A1] The original KD was proposed by Hinton in 2015. From then on, an intuitive sense is that larger teachers could teach students better because their accuracies are higher. In 2019, the research [1] first pointed out that the common sense is not right, first declaring that “the teacher accuracy is a poor predictor of the student’s performance”. Until now, this phenomenon is still **counter-intuitive**, **surprising** and **unexplored**.
>
> (a) the abstract in [2]: “…, a **counter-intuitive** argument is that better teachers do not make better students …”;
>
> (b) the introduction in [3]: “…, show **surprisingly** a student model distilled from a teacher with more parameters performs worse than the same one distilled from a smaller teacher …”;
>
> (c) the introduction in [4]: “… it remains relatively **unexplored** in knowledge distillation …”.
> Hence, exploring and explaining this counter-intuitive phenomenon is still valuable and promising.
>
> [Ref.1] On the efficacy of knowledge distillation. ICCV 2019.
>
> [Ref.2] Student Customized Knowledge Distillation: Bridging the Gap Between Student and Teacher. ICCV 2019.
>
> [Ref.3] Improved Knowledge Distillation via Teacher Assistant. AAAI 2020.
>
> [Ref.4] ResKD: Residual-Guided Knowledge Distillation. TIP 2021.
>
> [Q2] In Section 3, the teacher-student framework within which the authors are working is not carefully set up.
>
> [A2] Line 80 declares that $\mathbf{p}$ is the probability vector generated via softmax. Equation. (1) clearly shows the learning objective of the student, where $\mathbf{p}^T(\tau)$ is the teacher’s probability vector, which varies among different teachers (line 94).
>
> [Q3] The definition of $E_{j\neq y}$.
>
> [A3] We use $ E\_{j\neq y}[\mathbf{p}\_j^T(\tau)] = \frac{1}{C-1}\sum\_{j\neq y} \mathbf{p}^T\_{j}(\tau)$ to represent the average probability of classes without considering the correct class. $\mathbf{p}^T(\tau)$ is a softened probability vector with the temperatre $\tau$, and $j$ is the index of class. I think the formula of $ E_{j\neq y}[\mathbf{p}_j^T(\tau)]$ is easy to understand.
>
> [Q4] The doubt on $\mathbb{E}[\mathbf{p}]$ and $\mathbb{V}[\mathbf{p}]$.
>
> [A4] Although the notations are confused with the commonly utilized expectation and variance of a random variable, we only use $\mathbb{E}$ and $\mathbb{V}$ to denote the mean and variance of the elements in a vector for convenience. We use $\mathbb{E}[\mathbf{p}]$ and $\mathbb{V}[\mathbf{p}]$ to denote the mean and variance of a vector $\mathbf{p}$. We also clearly present the definition of $\mathbb{E}[\mathbf{q}]$ and $\mathbb{V}[\mathbf{q}]$ in line 121. **To eliminate the misunderstanding caused by the formula notations, we will replace $\mathbb{E}[\mathbf{p}]$ and $\mathbb{V}[\mathbf{p}]$ with $e(\mathbf{p})$ and $v(\mathbf{p})$, respectively.** (This may be viewed in the rebuttal revision.)
>
> $ e(\mathbf{p}) = \frac{1}{C}\sum_{j=1}^C \mathbf{p}_j$
>
> $v(\mathbf{p}) = \frac{1}{C}\sum_{j=1}^C \left(\mathbf{p}_j - e(\mathbf{p})\right)^2$
>
> **We will also replace $\mathbb{E}[\mathbf{q}]$ and $\mathbb{V}[\mathbf{q}]$ with $e(\mathbf{q})$ and $v(\mathbf{q})$, respectively. $\mathbf{q}$ is the vector after removing $\mathbf{p}_y$ from $\mathbf{p}$.**
>
> $e(\mathbf{q}) = \frac{1}{C-1}\sum_{j=1, j\neq y}^C \mathbf{p}_j$
>
> $v(\mathbf{q}) = \frac{1}{C-1}\sum_{j=1, j\neq y}^C \left(\mathbf{p}_j - e(\mathbf{q})\right)^2$
>
> [Q5] How the idea of asymmetric temperatures affects the input-output map of the teacher network?
>
> [A5] The KD process does not update the teacher network, and the teacher’s output is only utilized to supervise the student network. Hence, applying different temperatures does not affect the input-output map of the teacher.

---

> > ### Comment · Reviewer_RAjp · 2022-08-08
> > **Re: author responses**
> >
> > I thank the authors for their response and for modifying their draft. I have studied the paper again. It is still my assessment that it could be written far more clearly and with greater attention to detail, and that this would benefit the dissemination of this work enormously. At this point, however, I think that the paper makes a reasonable contribution that will be of interest to the KD community and that the reasons to accept the paper outweigh the reasons to reject. I am therefore increasing my score.
> >
> > What follows is a non-exhaustive list of suggestions that may help improve readability, and a few questions.
> >
> > The authors’ answer to my first question is a good deal clearer than the paragraph in which the idea that large teachers ought to teach better is introduced in the paper (lines 28-33). It would not hurt to explicitly state in the text that the reason larger teachers are expected to teach better is that they have the capacity to be more accurate.
> >
> > The teacher-student framework is still not clearly set up in the introduction. It would not hurt to explicitly state that the first step in a KD problem is to train the teacher network, and the second is to train the student network on the same training data by minimizing the two-part loss function given in Eq. 1. This type of simple contextualization would make the paper easily accessible not just to experts in KD but also to the larger ML community.
> >
> > Is there a typo in Eq. 1? Should $p_y^{S}(1)$ in the first term be $p_y^S(\tau)$? The authors mention in the appendix that they often set the temperature of the student network to $1$ but in Eq.1 as written, the student has two different temperatures - $1$ and $\tau$.
> >
> > I suggest that the symbol $E_{j\neq y}$ be replaced by $e_{j\neq y}$ for consistency with the new notation in the paper.
> >
> > The content of Lemma 4.1 is the simple fact that in the high temperature limit, the Boltzmann distribution over a discrete set of states approaches the discrete uniform distribution. I appreciate that a proof of this is provided. It would also be useful to explicitly state this sentence or something like it in the text. No reference to the uniform distribution appears in the text, but I think (please correct me if I am wrong) that the main idea behind ATS is simply that by bringing $p_y$ down closer to $1/C$ a little, one can create the wiggle room needed to push the distribution over wrong classes further away from the uniform distribution, and that this provides a better teaching signal for the student. If this is correct, then it may be useful to explicitly say something like it early on in the paper.
> >
> > In Corollary 4.7, are the temperature parameters of the two teachers the same?
> >
> > I suggest that a short discussion of how $\tau_1$ and $\tau_2$ should be chosen/are chosen in practice be included in the main text.
> >
> > In the appendix, it would be helpful to the reader if the names of the lemmas etc matched those used in the main text. For instance, the proof of Lemma B.1 in the appendix is the proof of Lemma 4.1 in the text.
> >
> > In Eq.3, I think the index $c^{\prime}$ should just be $c$. As written, the equation is inconsistent, and the free index is a dummy index anyway with the only constraint being that it cannot equal $y$.
> >
> > Lastly, I’d like to mention that I really appreciate the large number of experiments the authors ran and the many figures they included in the paper.

---

> > > ### Author Response · Authors · 2022-08-09
> > > **Responses to Reviewer RAjp**
> > >
> > > Thanks for your re-examination of our work. We are delighted that the reviewer could find our paper's advantages. We have carefully read the reviewer's suggestions and added some of the them to the rebuttal revision. The modified contents are shown in blue. The following will answer some questions proposed by the reviewer.
> > >
> > > [Q1] Make the paper clearer. Line 28-33 for the motivation. The teacher-student framework in Sect. 3.
> > >
> > > [A1] Thanks for the suggestions! We have modified the corresponding contents in the rebuttal revision. Please read lines 26-30 and lines 87-93. Additionally, we also add explanations for some basic notations, e.g., line 83 and line 128.
> > >
> > > [Q2] Typo in Eq.(1).
> > >
> > > [A2] When teaching the student, the loss contains both CE loss and KD loss. CE loss is the normal classification loss. KD loss transfers the knowledge to student. Commonly, CE loss takes a temperature of 1.0, while KD loss takes a temperature of $\tau$. For the student, its temperature could also be 1.0 in the KD loss. The formulation could be found in several KD works.
> > >
> > > (a) Equation (1) in Similarity-Preserving Knowledge Distillation. ICCV 2019.
> > >
> > > (b) Equation (3) in Improved knowledge distillation via teacher assistant. AAAI 2020.
> > >
> > > (c) Equation (20) in Contrastive Representation Distillation. ICLR 2020.
> > >
> > > In implementations, we set the temperature as 1.0 for CE loss. We set the temperature as 1.0 or the teacher's one for KD loss.
> > >
> > > These are further explained in lines 91-93 in the rebuttal revision.
> > >
> > > [Q3] The symbol of $E_{j\neq y}$.
> > >
> > > [A3] We replace it with $e(\mathbf{q})$ in the rebuttal revision and add definitions. Please see lines 106-108.
> > >
> > > [Q4] The content of Lemma 4.1 and the relation to the uniform distribution.
> > >
> > > [A4] In lines 139-140, we add the discussion about the uniform distribution when taking a higher temperature. However, we aim to ONLY analyze the statistics of WRONG class probabilities, where the mean and variance are more intuitive to analyze and calculate.
> > >
> > > [Q5] The main idea behind ATS.
> > >
> > > [A5] Thanks for the reviewer's comments. I think these comments are really useful to make our work more understandable. We add similar explanations in lines 207-208.
> > >
> > > [Q6] Corollary 4.7, are the temperature parameters of the two teachers the same?
> > >
> > > [A6] Yes, we keep the temperature the same, so we can analyze the difference between them. Notably, Remark 4.5 and Remark 4.7 both fix the temperature.
> > >
> > > [Q7] A short discussion of how two temperatures should be chosen/are chosen.
> > >
> > > [A7] We have investigated an ablation study in the original paper (Fig. 10). We vary two temperatures and empirically provide the chosen method of two temperatures in line 300. Notably, the results are reported based on THREE pairs of experiments (i.e., the three subfigures in Fig. 10), which may be more convincing than a SINGLE pair of ablation study.
> > >
> > > [Q8] Lemma B.1 in the appendix is the proof of Lemma 4.1.
> > >
> > > [A8] This is automatically generated via the LaTex. To solve this, we add the body references in Appendix to show them clearly. Please see line 549, line 561, and line 567 in Appendix.
> > >
> > > [Q9] The index of $c^\prime$ should just be $c$.
> > >
> > > [A9] Thanks for the reviewer's finding. This is a mistake. We have corrected it in the rebuttal revision. We replace $c$ with $c^\prime$ to denote the index of $\mathbf{g}$ and $\mathbf{q}$. We use $c$ to denote the index of $\mathbf{f}$ and $\mathbf{p}$. The latter two are vectors whose lengths are $C$, while the former ones have lengths $C-1$. The relation of $c^\prime$ and $c$ is: $c^\prime = c$ when $c < y$, and $c^\prime = c - 1$ when $c > y$. We also correct the notations in the Appendix Proposition B.3 and add some explanations.

---

### Author Response · Authors · 2022-08-02
**Responses to All Reviewers**

We would like to thank all the reviewers for their very helpful suggestions and thank the reviewers for their hard work. In the revised version of the article, we focused on revising the ambiguity in the mathematical symbol notations proposed by reviewer RAjp, hoping to solve the reviewer’s questions and make our contributions easier to understand. Thanks again to the reviewers!

---

### Author Response · Authors · 2022-08-09
**Rebuttal Revision Explanations to all Reviewers**

Thanks for the reviewer's work and valuable suggestions again! We provide the details for the newly-modified version. The modified contents are shown in blue.

(1) We take the reviewer RAjp's suggestions and re-organize the motivation and the background of the teacher-student framework in lines 26-30 and lines 87-93. We add some definitions for some notations in line 83, line 128, and lines 106-108. We add some discussions about the theoretical analysis and ATS in lines 139-140 and lines 207-208. A mistake of the index $c$ and $c^\prime$ is corrected in line 149.

(2) We take the reviewer iTrh's suggestions and add some additional comparisons with RE-KD and ReviewKD in Appendix Tab. 6 (lines 713-715). Discussions of the first term in ATS are also added in Appendix lines 543-547.

(3) We take the reviewer HVhD's suggestions and add some experimental details in lines 647-650, and lines 671-673.

---

### Meta-Review · Area_Chair_UGKC · 2022-08-21

**Recommendation:** Accept
**Confidence:** Less certain

**Metareview:**

This paper studies distillation consequences of carefully tuning the softmax temperature.  Reviewers found a variety of weaknesses but were positive overall.  I recommend acceptance, but that the authors carefully address the concerns raised during review, which will greatly improve the strength of the paper's message.

**Award:**

No

---

### Decision · Program_Chairs · 2022-09-14

Accept